# Work-Related Health Burdens of Nurses in Germany: A Qualitative Interview Study in Different Care Settings

**DOI:** 10.3390/healthcare10020375

**Published:** 2022-02-15

**Authors:** Jasmin Lützerath, Hannah Bleier, Andrea Schaller

**Affiliations:** 1Working Group Physical Activity-Related Prevention Research, Institute of Movement Therapy and Movement-Oriented Prevention and Rehabilitation, German Sport University Cologne, 50933 Cologne, Germany; h.bleier@dshs-koeln.de (H.B.); a.schaller@dshs-koeln.de (A.S.); 2Institute for Occupational Health Promotion, 50667 Cologne, Germany

**Keywords:** nurses, health burdens, work-related, workplace violence, setting-specific approach, qualitative research

## Abstract

*Background*: The growing need for nursing care is offset by a shortage of nurses, who are exposed to high physical and psychological demands in their daily work and have above-average absences that vary between different care settings. Based on the data on sick days, the question arises: What are the subjective work-related health burdens of nurses in acute care hospitals, inpatient care facilities, and outpatient care services? *Methods*: Sixteen semi-structured interviews were conducted in different care settings between May and September 2021. Questions about the professional career, everyday working life and personal health, violence in the workplace, and organizational framework conditions were asked. *Results*: The experiences of the nurses highlighted that health-related burdens have varying manifestations in different care settings. This was reflected in three main categories: health, everyday working life, and experiences of violence. In particular, the different structural framework conditions that trigger stress and the occurrence of violence are important from the perspective of the nurses. *Conclusions*: The results of this exploratory study can serve as a baseline for obtaining further setting-specific quantitative data that can contribute to the development, implementation, and evaluation of target group-specific health promotion programs.

## 1. Introduction

The increase in the need for care with a concurrent shortage of skilled nursing staff is currently the subject of heated discussions. In 2019, more than 19.4 million cases of acute medical health care were provided to patients in German acute care hospitals [1]. It is assumed that the number of treatment cases in acute care hospitals will continue to increase in the coming years [2] as the risk of disease and need for care increases in the aging population [3,4]. Additionally, more than 4.1 million people in Germany are currently in need of long-term care [5]. According to the social code XI §14, which defines the term “need for care” for the purpose of receiving benefits from the statutory long-term care insurance, persons are entitled if they have health-related impairments of their autonomy or abilities and therefore require help from others [6]. Of these persons, nearly one in five receive permanent (19.3%) or temporary (0.5%) care in inpatient care facilities, and 23.8% use outpatient care services [5]. People in need of long-term care can live permanently in inpatient care facilities that provide accommodation, catering, nursing care, and social support [7]. In addition to household tasks, outpatient care services dispense both long-term and treatment care, which is provided when acute health care is needed regardless of the reason for long-term care, whereby these are booked individually and partly reimbursed by the statutory insurances [8,9]. The possibility of receiving treatment care through outpatient care services was used in 2.6 million cases in 2019 [10]. This goes along with the fact that the tasks and demands on nurses vary in the different care settings [11]. In recent years, the demand for nurses has continued to be higher than the number of available staff on the labor market [12,13,14]. Until 1 January 2020, there were two types of nursing apprenticeships in Germany to enter the market as a professional nurse: health care nursing, trained in acute care hospitals and outpatient care services, and geriatric care nursing, trained in geriatric inpatient care facilities or outpatient care services for the elderly [15,16]. Nursing care is currently provided by an increasing number of around 1.7 million nurses in Germany with around 1.1 million health care nurses and 615,000 geriatric care nurses [12], in a given mix of professions between the care settings [5]. Of these, approximately 458,000 nurses work in acute care hospitals [1], around 796,000 in inpatient care facilities, and 422,000 in outpatient care services [17].

In general, although the health burdens in nursing are well known, in-depth evidence on subjective work-related differences between settings is lacking. Job demands can lead to a health burden when they exceed employees’ capabilities [18], which in turn can be a predictor of absenteeism when combined with work resources [19]. For years, nurses have had higher sick leave rates (24.8–25.5 days/year) than employees in other professions (18.8–19.5 days/year) [20]. The most frequent physician-diagnosed diseases of nurses are respiratory diseases, musculoskeletal disorders, and psychological impairments, with geriatric care nurses having more and longer sick leave than health care nurses [21,22,23]. Overall, one in four nurses rate their ability to work as moderate or poor [24], although there appear to be differences in relation to the care setting. Workability is rated worse in acute care hospitals than in inpatient care facilities and best in outpatient care services [25]. According to the available German data, while nurses across settings seem to be chronically stressed, with the highest stress levels found in nurses working in acute care hospitals [26], it seems as if there are hardly any differences regarding their physical health or well-being [26,27,28]. However, outpatient care nurses appear to be more frequently affected by stress symptoms and their consequences, such as burnout, than nurses in inpatient care [21,22,23].

The current literature thus indicates that there might be differences in the work-related health burdens of nurses according to the care setting. It is assumed that knowledge of setting-specific differences may be central to health systems’ efforts to maintain a healthy workforce. Therefore, the present study aimed to gain insights into the subjectively experienced work-related health burdens of nurses in different care settings. The research question derived from this was: What are the subjective work-related health burdens of nurses in acute care hospitals, inpatient care facilities, and outpatient care services?

## 2. Materials and Methods

This qualitative study was designed based on the criteria for reporting qualitative research (COREQ) [29] and was conducted as part of a broader project: “Workplace offers for health promotion and violence prevention” (BAGGer), funded by the German Federal Ministry of Health (BMG) from 2020 to 2022. The project aims to implement and evaluate workplace health promotion for employees in care settings who are exposed to particular physical or psychological demands. This study belongs to the first project phase (A): Identification of relevant health problems. This project is registered in the German Clinical Trial Register (DRKS00024961).

### 2.1. Participants

Participants were included in the study if they fulfilled the following criteria: (1) they are a professional nurse; (2) a minimum age of 18 years; (3) working in an acute care hospital, inpatient care facility, or outpatient care service. The exclusion criteria were: (1) professional nurses mainly assigned to administrative working tasks; (2) apprentices. The recruitment of the participants was conducted through the persons in charge of employee health at the participating employers of the BAGGer-project. In this process, nurses could volunteer to participate by contacting the person in charge, and thus the selection was random. The first contact to the participants was by e-mail, to generally inform them about the research project and to request an interview appointment. All selected persons agreed to participate and written informed consent was provided by the participants.

The total sample consisted of four men and twelve women with an age range from 25 to 54 (mean 39 ± 11). In total, seven of the interviewed nurses had higher qualifications e.g., as a student instructor or in a special nursing field. The nurses had between 4 to 36 years (mean 14 ± 9 years) of experience in the care sector and were working as professionals in their given settings, where they were interviewed from 3 months to 15 years (mean 6 ± 4 years). Three nurses had a migration background. In every setting, one person was working part-time. The interviews lasted between 36 to 171 min (mean 78 ± 33 min). A more detailed list of the participants by setting can be found in Table 1.

### 2.2. Interview Guide

A problem-centered interview guide was developed to structure the interviews [30]. The questions were collected and formulated collaboratively by the research team to identify potential health burdens of nurses in different care settings. The interview guide contained open-ended questions and was divided into five different topic blocks with key questions or varying follow-up questions oriented to the problem to be examined in a flexible order. Table 2 shows the main topics (professional career, everyday working life, health, workplace violence, company) and key questions of the interview guide. 

To ensure comprehensibility and to estimate the amount of time required for the interviews, the guide was pre-tested with a nurse working in an acute care hospital.

### 2.3. Data Collection

The interviews were conducted in the period between May to September 2021 by two research team members, both of whom were Ph.D. candidates and trained in qualitative research. Due to the rapidly changing visitation rules in care settings as a result of the ongoing COVID-19 pandemic, telephone interviews were arranged with the interviewees. The participants were either at home or had a generous amount of time at work for the interview in a quiet room, so that only the researchers could hear their words and audio-record them. The interviewees did not know the researchers and were simply informed that they were interested in workplace topics. The demographic data and field notes were completed after the interview as to not disturb the flow of the discussion. After the first fifteen interviews, five per setting, the data saturation was discussed. Since new health-related issues were brought up in the setting of inpatient care that promoted understanding, the researchers requested contacts to a further interview partner.

### 2.4. Data Analysis

The transcription of the interview recordings was done by a professional typist according to Dresing-Pehl [31]. The anonymized transcripts were analyzed according to the method of structuring content analysis, which is popular in German-speaking countries [32,33]. The method is comparable with the framework method to analyze qualitative data [34]. An initial coding system was developed deductively based on the main topics and content issues of the interview guide as seen in Table 2. An initial sample of five transcripts was coded and critically (re)read to confirm the main categories based on the interview guide. This was followed by several analysis steps to inductively formulate further categories and subcategories. The codes and text passages were discussed and compared by the research team before the entire dataset was finally coded by three researchers using MAXQDA Standard 2020 software, by VERBI GmbH Berlin. To analyze the data, grid summaries were created to produce summary tables, linking the codes and cases to compare the setting groups.

## 3. Results

Concerning the research question, the following main categories were deduced: (a) health, (b) everyday working life, (c) experiences of violence. Each main category was divided into subcategories showing different characteristics (see Table 3). 

In the following, the results of the content analysis of the statements of nurses from acute care hospitals (ACH), an inpatient care facility (ICF), and an outpatient care service (OCS) are presented in relation to the respective main and subcategories.

### 3.1. Health

The main category “health” comprised statements relating to health and health behavior. Three subcategories were identified: health status, private health behavior, and health behavior at work. 

The health status subcategory was defined as the subjective assessment of physical, mental, and social health. The statements indicated that most of the interviewed nurses considered their state of health to be good despite some physical or social burden factors. The physical burdens mentioned were back pain, head and neck pain, problems with the knees, legs, and feet, or hypertension. It was noticeable that these burdens were predominantly named by OCS and ICF nurses. In contrast, mental health problems were mentioned equally in all three settings. The majority of the nurses reported feeling mentally or emotionally exhausted, as shown by the difficulty they have had in switching off after work, frequently feeling irritable, being listless, or already being treated for a mental health disorder. Likewise, in all three settings, most nurses reported frequently being tired, which for some meant that they hardly participated in social life. In ACH and OCS, some already spoke of chronic sleep problems in the form of difficulty falling asleep and staying asleep.

The subcategory of private health behavior was defined as health-related behaviors in private life. Across all settings, nurses reported promoting their health through relaxation, especially sleeping. Additionally, social exchanges about work experiences was considered particularly relaxing. 


*“Sure, I talk as well. You always have to be careful how much you talk in a private setting. But, with particular events you talk things over with friends, maybe go over things once more.“*

*(ACH nurse)*


Active relaxation, for example by doing sports, was generally described as beneficial. However, most of the nurses interviewed reported that this was not possible for them due to time constraints or exhaustion.

The third subcategory, health behavior at work, was defined as the individual’s ability to behave in a healthy manner at work. The nurses in all three settings reported specific challenges in dealing with stress. They could not always deal with it appropriately, which manifested itself in neglecting some nursing tasks or making mistakes in performing them. Across all settings, it was frequently reported that breaks, even when given, were not taken due to time pressure. Several ACH and ICF nurses also mentioned smoking or eating an excessive amount of sweets to experience a moment of peace amidst the many demands they face at work. While all nurses were familiar with ergonomic working methods, mainly nurses suffering from lower back pain reported working ergonomically. However, most of them mentioned that they neglect ergonomic ways of working when they are under time pressure and colleagues or equipment are not immediately available, or when they feel pressure to perform in emergency situations. In OCS, the main nutrition issue addressed was that they drink little to avoid having to go to the toilet, and often eat while driving or on the side of the road. Particularly in ACH and ICF settings, the nurses interviewed stated that their nutrition during work hours was poor. Especially when they are under time pressure, they eat quickly or not at all, and they rarely take the opportunity to drink.

### 3.2. Everyday Working Life 

The main category “everyday working life” comprised structural framework conditions at the workplace and was divided into six subcategories: working hours, quantitative demands, work environment, qualitative demands, job control, and working atmosphere.

The subcategory working hours described the period during which the nurses carry out their professional tasks. Weekend work and atypical working hours, i.e., starting work before seven in the morning, were reported in all settings. Both were perceived as stressors, especially by interviewees with families. Another characteristic, mainly reported by ACH and ICF nurses, was the night shift, which affected personal sleep and eating patterns. 


*“Because I work in care, I’ve been on night duty for years on a rotating shift system. My biorhythm is completely messed up then. When you’re off, you wake up at five in the morning anyway, you’re wide awake, you’re driven, your eating habits are completely changed.”*

*(ACH nurse)*


All interviewed nurses referred to the fact that a regular change between shifts might lead to shortened rest periods, which was perceived as physically demanding. Additionally, overtime on a regular basis and working more than seven days in a row were named as stress factors and were negatively associated with individual workability.

The subcategory quantitative demands was defined as the perceived amount of work in performing the job. Tasks that are distant from the patients, such as documentation, organization, service, and cleaning activities, were mentioned as forming part of a steadily increasing workload over the years, and these have once again increased sharply due to the pandemic. While OCS nurses perceive a reduction in workload due to digitization, ACHs, on the other hand, reported that tasks on the computer have increased due to the rise in patient turnover, and that the time is then lacking for the patients. Another stressor primarily mentioned in ICF and ACH was frequent patient interaction. In addition, the growth in the number of multimorbid patients in need of care was described in all settings, making the nursing and care duties increasingly complex and time consuming. Furthermore, the patient volume has increased, while at the same time there are fewer colleagues available, especially in ACH and OCS. In ICF, in contrast, a more differentiated picture emerged. On the one hand, the number of nursing staff and assistants has increased, while the number of patients has remained constant; on the other hand, it was also emphasized that the staffing ratio is still not sufficient to feel any real physical relief.

The third subcategory, work environment, was defined as the physical conditions of everyday work. The spatial environment was characterized by long walking distances in all settings, which in OCS was even more physically demanding because of needing to constantly climb stairs. While ACH and ICF spoke of a lack of private spaces on the wards, OCS nurses, in contrast, sometimes worked in cramped private residences, which was a recurring physical burden, and those needing care do not always have clean homes, which sometimes causes disgust, making toilets unusable. In addition, OCS nurses spent a great deal of time in their cars, which were only partially adapted to their own physiological needs, and are exposed to weather, road, and traffic conditions, and which can cause time delays. Work equipment that facilitates ergonomic work was described by some in the ACH and ICF as being distantly located, outdated, or not sufficiently available. In OCS, patients are responsible for providing both equipment (such as lifts) and work materials for treatment care, which regularly results in work equipment not being fully available.


*“Especially in care jobs today, you simply have to improvise […]. If you have to place a catheter, you are happy when you have dressings, disinfectant or even a urine bag and sterile gloves in the package.” *

*(OCS nurse)*


The nurses reported that they were also exposed to physical hazards such as bacterial or viral sources of infection in the course of their daily duties, which means that they then have increased contact with disinfectants and constantly work with personal protective equipment, which is perceived as a physical burden, especially on warm days.


*“That I have trouble breathing, I often have headaches, well, headaches, dizziness, circulation problems; this happens regularly since we have had to mask up.”*

*(ACH nurse)*


The subcategory of qualitative demands was defined as the emotional and cognitive work demands that are experienced. Dealing with suffering, e.g., due to illness or death of patients, was described as emotionally stressful, especially in the first years of the job and when exposed to the situation alone without colleagues on site, such as during night shifts or in OCS. Especially ICF and OCS nurses mentioned that observing the suffering of care receivers due to a lack of financial means also triggers compassion in them, and they worry about their own financial status in old age.


*“Or when I hear that they have to go to the food bank. Seriously ill people, dependent on oxygen, and have so little money that they have to go to the food bank. Well, these are the things which break our spirit more than the work; How the elderly have to live.”*

*(OCS nurse)*


Most nurses indicated that time-intensive interaction work with patients goes a long way toward providing care professionally and in line with their own moral values, which, especially for ACH and ICF nurses, also contributes to patients seeking contact less frequently. In OCS, on the other hand, it was referred to as burdensome that interaction work is a service for which nurses are rarely booked, and therefore does not show up in their planned schedule. Furthermore, the responsibility for the correctly performed nursing care was described as stressful, especially when they are the only trained nurses on-site and, in addition, as in ICF and OCS, there is often no physician on-site.


*“So yes, when there is something wrong with a patient, I have to decide within seconds: “What must I do? Do I need a doctor, or should I just call the relatives so that they can deal with this? Or do I have to call an ambulance?” […], creates stress somehow, the stress levels rise. You really notice it.”*

*(OCS nurse)*


The job control subcategory was defined as the ability to shape or plan one’s workday. The workflow, especially in ACH and ICF, is characterized by frequent work and break interruptions by other patients or relatives, which means that multiple tasks have to be completed in parallel.


*“I have sometimes experienced that I start several things, but then I have to abandon them. Because between those things that you already have to deal with, something else comes between. It is very draining to have to keep an overview of things. And, of course, you go home with a heavy head, and with the feeling that you have forgotten something, or needed to do something or needed to do it better.”*

*(ICF nurse)*


The assignment of work in ACH and ICF settings is mostly done collaboratively within the care team, with some tasks being added during the course of the shift. In contrast, OCS nurses report that the work plan and any changes are meticulously provided by their superiors. Many nurses stated that they constantly have the feeling that they are on unpaid on-call duty to take over a colleague’s shift at short notice, e.g., if the colleague is absent due to illness. There were also contrary statements concerning regular calls where, in the case of absences, no one is asked to step in, and the workload is then distributed among those present.

The sixth subcategory, working atmosphere, was defined as the experienced work climate in terms of cooperative work, sense of community, and rewards. In ACH, cooperation with physicians is sometimes conflictual due to a strong hierarchical gradient, while ICF and OCS experience it as exhausting to get in touch with them at all. In OCS, prescribed medications and nursing equipment are often delivered by pharmacies or medical supply stores. In this context, many of the interviewees reported that there are always long delivery times, which means that they often have to do without the equipment that they need. In addition, especially ACH and ICF nurses reported that language barriers result in missing information and conflicts with service or assistance staff. It was noted in all settings that information is often lost, especially during external patient transfers, and that this leads to extra work and incorrect treatment approaches. While a high level of cooperation is often observed in the nursing teams, the sense of community in the overall organization is experienced as low, especially in ACH and ICF, when cost pressure or pressure to perform is passed on by top management.


*“Even so, you have to work more effectively. Then there is a discussion about consumption of materials, and I think: Please, if somewhere material costs should not be discussed about, then it is in the medical field.“*

*(ICF nurse)*


The awareness of doing something useful for society and experiencing gratitude was frequently mentioned as rewards for daily work. In contrast, there is a low experienced appreciation from society, also in financial terms, which in turn limits the financial possibilities for health care expenditures.

### 3.3. Experiences of Violence 

The main category “experiences of violence” includes statements about experiencing violence in the workplace. Three subcategories were identified: understanding of violence, forms of violence, and consequences.

The subcategory understanding of violence was defined by what nurses understand under the term “violence” and their individual role in this context. Most nurses often experienced themselves in the role of the persecutor of violence. In this context, they defined it as violence if they cannot take care of the patient in a needs-oriented way due to the framework conditions of their daily work, or if they were tired and stressed and therefore reacted irritably to the patients.


*“Violence in care work for me is, of course, when it comes to scuffles, to physical assaults. But also, if you are markedly distanced towards patients and ignore them, that for me also is violence, in particular if it means ignoring the wishes or needs of the patients.”*

*(ICF nurse)*


In terms of the nurses themselves being victims of violence, this was mostly downplayed and excused by patients’ illnesses or cognitive limitations. It was mentioned that it might be part of the nursing job to fulfill the expectations of patients and relatives regarding performance or time. In addition, nurses perceived themselves in the role of a witnessing rescuer when patients attack each other, as seen in ACH and even more so in ICF. Also, the sense of having to rescue a patient occurs in ACH e.g., when relatives with a health care proxy request physical intervention against the patient’s will, which is difficult for nurses to reconcile with their own morality, and in OCS, when relatives are overwhelmed with care and use violence.


*“There is much more propensity for violence; I don’t know why. But the inhibition threshold of patients or their relatives is so lowered that sometimes I really, I find it scary.”*

*(ACH nurse)*


The second subcategory, forms of violence, was defined as verbal, physical, and sexual experiences of violence and their subjectively perceived frequency. 

Verbal violence in the form of insults, verbal abuse, or bossing around by patients and relatives was experienced daily. Physical violence, for example, hitting, pinching, kicking, or holding was also mentioned. It was striking that, on the one hand, this was often described in questions about everyday work but, on the other hand, most nurses stated that they rarely experienced physical violence. Sexual violence in the form of intimate touching was mentioned but mostly dismissed as being unintentional, especially in ICF and OCS. In contrast, in ACH, physical and sexual violence in the form of (non)verbal or physical harassment was sometimes experienced as intentional, which was perceived as threatening, especially when this occurred when the nurse is alone, e.g., during the night shift.

The subcategory consequence was defined as how the organization and persons handle incidents of violence. Especially in ICF and OCS, management and physicians are involved in discussing how such situations can be avoided through nursing measures or medication, whereas in ACH, discussions with psychologists were offered or no support was experienced from colleagues or the organization at all. Although most nurses reported that they personally cope well with violence, some indicated that they reflect on these incidents and feel guilty because of their own (re)actions, and in some cases, an uncomfortable feeling remains, and they try to avoid the patients and/or situations. Some nurses also indicated that they had nightmares and—to some extent—anxiety, especially when the incident was perceived as threatening and intentional, regardless of whether they received or were offered support.


*“And then the patient, well, he was undressed, he just tried to push me against the wall. What exactly he intended—I don’t know. Yes, that was such an event, it really stayed in my mind, because in that moment I was actually completely overwhelmed and I just didn’t know how I should somehow deal with that. And because even in hindsight, I somehow,—which is actually completely unnecessary—that’s also what my boss told me, because actually I felt guilty because I knocked the patient down.”*

*(ACH nurse)*


## 4. Discussion

This qualitative study aimed to determine the subjective work-related health burdens of nurses in ACH, ICF, and OCS. The results show a variety of burdens experienced by nursing staff in all three types of care settings. 

### 4.1. Summary of Findings

Above all, time and performance pressures were experienced as stress factors. In all settings, time pressure was experienced most notably when quantitative demands increase. ACH and ICF nurses named frequent patient contact as one of the causes. Furthermore, ACH nurses in particular stated that tasks away from patients are becoming more intense. Especially nurses in ACH and OCS mentioned that they were responsible for too many patients—a quantitative demand that triggered time pressure. The OCS nurses added that they have no influence on the road and traffic conditions, which can cause delays, and that they have hardly any influence on their work plan. Pressure to perform was also always associated with quantitative demands by the nurses interviewed, and it was indicated that time-intensive interaction work plays a major role. ACH and ICF nurses justified the lack of time for interaction work due to the quantitative demands of their work, while the OCS nurses highlighted the fact that these tasks do not appear in the work plan due to the remuneration system. In the settings of ACH and ICF, performance pressure also resulted from work interruptions. In this context, the interruptions resulted in many tasks being worked on in parallel and the nurses were not always sure whether they could, or had, fulfilled all of them as a result. Furthermore, ICF and OCS nurses in particular feel pressured because they often have the sole responsibility for patients, without support from other trained nurses or physicians. 

Our qualitative results on the subjective work-related health burdens are in line with the available quantitative results in the literature. According to these results, around two-thirds of nurses in Germany have been reporting for years that they are often under high time or performance pressure (59–67%) [35,36]. In addition, employees in the health and social care sector in particular have experienced increased demands due to pandemic-related extra workloads [37]. Further, nurses in Germany frequently suffer from musculoskeletal disorders and psychological impairments, with geriatric care nurses more commonly affected [21,22,23]. However, our qualitative results go beyond the current research state, as they provide a deeper insight into the subjective health burdens related to the nursing setting. Physical complaints were mainly mentioned by ICF and OCS nurses in our study. The main reason given for this was the lack of colleagues or work equipment to support the physical work on the patient such as lifting and carrying. In ICF, this was attributed to the nurse-to-patient ratio and long distances to equipment or insufficient equipment purchases by the employer. In OCS, on the other hand, it was common to work alone, and work equipment was unavailable if patients did not purchase it or if it was delivered late. In addition, OCS nurses often stated that they work in awkward positions because patients’ homes did not allow them to do otherwise, and their car could not be adapted to their own physical needs. The current quantitative data mainly focuses on profession-specific differences, rather than the setting-specific differences, in work-related health burdens. Against the background of different setting-specific working tasks, it must be assumed that geriatric care nurses experience frequent standing, lifting, and carrying of heavy loads, and working in awkward postures [36], which can lead to spinal disorders and lower back pain [38,39]. However, our results suggest that lifting and carrying heavy loads, and working in awkward postures was more an ICF and OCS nurses’ issue than a professional group one. This is consistent with findings from occupational safety and health research that more tasks are performed on patients in ICF and OCS than in ACH, where more coordinative tasks are involved [11]. Our results also indicate that in ACH and ICF, demarcation conflicts between nursing tasks on the one hand and service tasks, on the other hand, are mainly caused by the coordination of workflows with service or assistance staff, which in turn affects performance. An improvement of the cooperative collaboration between professionals and institutions is desirable to reduce the quantitative demands caused by a lack of information. Common research suggests that back and neck pain are more likely to be associated with psychosocial factors, such as quantitative demands [40]. The results also indicate that performance suffers under stress. In line with this, with the increasing workload, around one-third of nursing tasks on a shift are commonly not performed in Germany, with nurses first neglecting interaction work [41]. A lower level of interaction work, on the other hand, was reported as a predictor of increased quantitative demands in this study, especially in ACH and ICF. 

Emotional exhaustion was also a central theme for nurses in all settings. In ACH, about 70% are affected at least once a month, in ICF about 58%, and in OCS approximately every second nurse [27,28]. As mentioned, qualitative demands occur to varying extents in all settings and are described as particularly exhausting when nurses are exposed to them alone, as occurs during night shifts in ACH and ICF or predominantly in OCS. Emotional exhaustion is associated with qualitative demands and, moreover, with poorer psychological and physical well-being [42]. The in-depth results of this study provide insights into how this association could come about. Being too exhausted was also one of the central themes mentioned, which led to neglecting healthy behavior. This involved not engaging in social and sports activities in private, as well as neglecting ergonomic ways of working and taking breaks at work. In addition, nurses in ACH and ICF, in particular, reported that they frequently skip breaks due to the quantitative demands of their work. Considering the detailed qualitative statements, it is not surprising that quantitative studies also found a relationship between frequently skipped breaks and psychosomatic, as well as musculoskeletal, complaints [43]. Additionally, ACH and OCS nurses in particular reported sleep problems, which they often attributed to their work hours. This is hardly surprising since the majority of German nurses work weekends (85–87%) and more than half work nights or at atypical times whereby they have little control over their own working hours, which results in shortened rest periods (under 12 h) for one in two health care nurses and one in four geriatric care nurses [44]. Working atypical times and having a low level of decision latitude or job control is associated with outcomes such as cardiovascular diseases [45,46], depressive symptoms [47], and musculoskeletal complaints [48]. In this context, OCS nurses in particular stated that they had hardly any influence on their own workflow.

When compared to other professions, the nursing profession differs not only in terms of increased sick leave but also in terms of regular experiences of violence, which were described by the nurses as a typical work demand. Astonishingly, an everyday work demand such as the experience of violence has hardly been investigated in Germany [49]. In one year, 84–97% of nurses in Germany experience verbal attacks, most frequently in ACH, and 61–77% have experienced physical violence, especially in ACH and ICF [50,51], while in other jobs, violence, including collegial violence, is experienced by 16% of employees [52]. According to the results, nurses in ICF seem to be more affected by violence than in OCS [22,25]. In contrast, ICF and OCS nurses in particular denied having experienced physical violence, and sexual violence was mentioned more in passing. Consequently, it can be assumed that further sensitization to the topic of violence could be beneficial. From the data, it was shown that 69–75.9% of nurses in Germany, especially in ACH, are exposed to verbal sexual violence at least once in 12 months, whereas physical sexual violence is more likely to occur in ICF but is also experienced by at least every second nurse in other care settings [53]. The nurses interviewed indicated that the personal consequences of physical and sexual violence are experienced independently of the organizational support received. The meta-analyzed data from 41 studies found that the organizational handling of violence influences its prevalence, and it is known that frequent occurrences, especially of sexual violence, have a negative impact on mental health [54]. A personal consequence resulting from this that was mentioned is that situations or patients are avoided after an incident and it was often followed by feelings of guilt, particularly if the affected nurse felt that they had not acted according to their moral values. This is consistent with nurses’ experience of moral injury, which occurs when witnessing human suffering, boundary violations, or violence on multiple occasions [55]. In this study, ICF and OCS nurses in particular indicated that they were deeply affected by patient suffering. Boundary violations of their own moral ideals was especially highlighted by ACH and OCS nurses when relatives have a different idea of the necessary patient care than the nurses. Nurses in all settings also indicated that they act against their morals when they cannot provide care according to the patient’s needs, which can result from increasing quantitative demands, and is an act of violence from the perspective of the nurses. Another relationship exists between frequent moral injury and destructive behaviors and aggression toward others [56]. Since nurses indicate that their experiences of violence are becoming more frequent and that need-based care decreases in stressful situations, it is not unexpected that quantitative data would establish a correlation. Studies have shown that approximately one-third of nurses in Germany feel strongly stressed by incidents of violence [50,51]. Our qualitative data go a step further and show that not only do nurses feel stressed when they experience violence, but also that stress in the workday promotes the occurrence of violence.

### 4.2. Strength and Limitations

The results of our study provide a setting-specific insight into the work-related health burdens of nurses in Germany. Key aspects and differences are highlighted that need to be known to improve setting-specific working conditions for nurses. Despite these findings, there are also limitations in the study. For example, the participants were selected through contacts in care organizations participating in the health-promoting BAGGer project. Therefore, it might be assumed that the participants tended to work for care organizations that are interested in the health and wellbeing of their employees, and therefore also experience fewer work-related health burdens. Accordingly, the influence of the organizational level should also be examined more closely in the future. A complementary or subsequent step could be, for example, to interview directors and managing professionals and to contrast the results with the nurses’ statements. In addition, the inclusion criterion “professional nurse” was chosen, which only partially reflects the profession and skill mix of the sector. In acute health care, more than four out of five are trained nurses or have higher qualifications (12%), while in geriatric care, around half are trained nurses (52%) and the rest are nursing assistants [12]. Moreover, nurses with higher qualification levels are associated with better health outcomes [23]. Also, through this convenient selection, mainly full-time nurses were interviewed. In comparison, about half of the trained nurses work part-time [10] with part-time employees having the highest sickness-related absenteeism [21]. Furthermore, the interviews were conducted by telephone, which had the advantage that the participants were in a confidential environment, but a pure soundtrack contains less information about the interviewee than a face-to-face conversation. Although data saturation was discussed and no more new thematic issues (code saturation) were mentioned, it is questionable whether complete meaning saturation was achieved [57]. A major strength of the study presented is that, given the sensitive nature of the topic of violence, the interviewers had a psychological hotline available in case the topic might trigger the participants, which was not necessary in this case.

## 5. Conclusions

Maintaining nurses’ ability to work plays a central role in health care. This exploratory study shows that, from the subjective perspective of nurses, stress arising from varying demands in different settings takes on a prominent role in their everyday work. In this context, the experience of stress influences nurses’ own health behaviors as well as their interactions with patients, which has been linked to the occurrence of violence. To positively influence this with target group-specific health promotion programs, behavioral training for nurses, including raising awareness of violence, on the one hand, and reducing work demands in combination with employee-oriented changes in organizational culture, on the other, could be beneficial. Nevertheless, more setting-specific data are needed to quantify the work-related health burdens of nurses occurring in different settings. On this basis, setting-specific health promotion programs for nurses could be developed and implemented, or existing ones evaluated. We hope that this study helps to shape discourse on the relationships among setting-specific care work, incidents of violence in nursing, and nurses’ health.

## Figures and Tables

**Table 1 healthcare-10-00375-t001:** Sample description and interview duration by care setting.

	Acute Care Hospital (*n* = 5)	Inpatient Care Facility (*n* = 6)	Outpatient Care Service (*n* = 5)
Age [years] mean (±SD); minimum-maximum	33 (±6); 27–42	34 (±11); 25–52	51 (±3); 47–54
Gender: female [n; %]	3 (60%)	4 (67%)	5 (100%)
Profession: geriatric care nurses [n; %] higher qualification [n; %]	2 (40%) 1 (20%)	5 (83%) 1 (17%)	2 (40%) 5 (100%)
Experience in the care sector [years] mean (±SD); minimum-maximum	12 (±3); 4–16	11 (±7); 5–20	20 (±12); 4–36
Working as a professional in the setting [years] mean (±SD); minimum-maximum	7 (±4); 2–12	7 (±5); 0.25–15	4 (±3); 1–8
Migration background [n; %]	0 (0%)	1 (17%)	2 (40%)
Working part-time [n; %]	1 (20%)	1 (17%)	1 (20%)
Interview duration [minutes] mean (±SD); minimum-maximum	69 (±7); 59–87	63 (±30); 36–119	104 (±42); 58–171

SD = standard deviation; *n* = number.

**Table 2 healthcare-10-00375-t002:** Topics and key questions of the interview guide (translated from German).

Topic	Key Question
professional career	“How did you come to your profession?”
everyday working life	“What does a typical working day in nursing look like?”
health	“How do you define health for yourself personally?”
workplace violence	“What kind of conflicts or violence do you witness at work?”
company	“What do you think makes your care organization different from other employers?”

**Table 3 healthcare-10-00375-t003:** Coding system with defined main and subcategories and meaning units.

Main Category	Subcategories	Characteristics
Health	Health status	physical mental social
	Private health behavior	promote neglect
	Health behavior at work	stress management ergonomics nutrition
Everyday working life	Working hours	weekend atypical times night shifts short rest periods long duration
	Quantitative demands	tasks distant from patients frequent patient interaction time-intensive patient care nurse-to-patient-ratio
	Work environment	spatial environment work equipment physical hazards
	Qualitative demands	dealing with suffering interaction work responsibility
	Job control	workflow assignment of work on-call duty
	Working atmosphere	cooperative work sense of community reward
Experiences of violence	Understanding of violence	persecutor victim rescuer
	Forms of violence	verbal physical sexual
	Consequences	organizational personal

## Data Availability

The data presented in this study are available on request from the corresponding author.

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
