# Peer review of "Work-Related Health Burdens of Nurses in Germany: A Qualitative Interview Study in Different Care Settings"

_healthcare, 2022, doi:10.3390/healthcare10020375_

Round 1

Reviewer 1 Report

I have read the manuscript " Subjective work-related health burdens of nurses – a qualitative interview study among nurses working in acute care hospitals, inpatient care facilities and outpatient care services in Germany " with care and interest and have the following observations, queries, recommendations:

1) The manuscript deals with an important issue and thus will have great traction with the broad readership.

2) The title is too long in its present form; therefore, the title needs modification. Please include the most relevant and novel part of the manuscript in the title, making the title concise and precise.

3) The background of the study and literature review should be updated to include the most recent and important works published in MDPI journals and other key journals. Some of the following studies could be a useful guide for the authors:

https://doi.org/10.1111/jep.13611; https://doi.org/10.3390/ijerph17082954; https://doi.org/10.3390/ijerph18094577; https://doi.org/10.1016/j.envres.2020.109995; https://doi.org/10.1007/s11869-020-00943-2 

4) The starting paragraph of the introduction is too short (2 lines). Please merge it with the second paragraph. Avoid the same situation throughout the manuscript.

5) The novelty statement seems weak in its present form. Please improve the explanation of contributions made by this paper.

6) The selection of participants was random or did the authors use a convenience sampling technique? Needs clarification.

7) Where were the scale items of constructs borrowed or modified from? Provide sources.

8) Please avoid heading after heading with nothing in between. Instead, provide at least a small paragraph to state the contents under each heading and sub-headings.

9) Conclusions are in good order.

Author Response

Thank you very much for your observations, queries and recommendations. Please find our point-by-point answers in the attached document. Your comments helped us a lot to substantially improve the quality of our manuscript.

Changes were marked up in the manuscript using the “Track Changes” function.

Response to Reviewer 1 Comments

Point 1: The manuscript deals with an important issue and thus will have great traction with the broad readership.

Response 1: Thank you very much for pointing out the potential of our manuscript and the opportunity to revise it based on your comments. Your comments helped us a lot to substantially improve the quality of our manuscript.

Point 2: The title is too long in its present form; therefore, the title needs modification. Please include the most relevant and novel part of the manuscript in the title, making the title concise and precise.

Response 2: We have now changed the title to "Work-related health burdens of nurses – a qualitative interview study among nurses working in acute care hospitals, inpatient care facilities and outpatient care services in Germany”

Point 3: The background of the study and literature review should be updated to include the most recent and important works published in MDPI journals and other key journals. Some of the following studies could be a useful guide for the authors:

https://doi.org/10.1111/jep.13611; https://doi.org/10.3390/ijerph17082954;  https://doi.org/10.3390/ijerph18094577; https://doi.org/10.1016/j.envres.2020.109995; https://doi.org/10.1007/s11869-020-00943-2  

Response 3: Many thanks for this advice and the references. Since our results are limited to the German health care system, we prefer to limit ourselves to a German reference for this statement because of the international differences in health care systems. We have highlighted this as follows:

“According to the available German data, while nurses across settings seem to be chronically stressed, with the highest stress levels found in nurses working in acute care hospitals [26], it seems as if there are hardly any differences regarding their physical health or well-being [26–28].”

Point 4: The starting paragraph of the introduction is too short (2 lines). Please merge it with the second paragraph. Avoid the same situation throughout the manuscript.

Response 4: We have merged the first paragraph with the second one and carefully checked paragraph lengths in the manuscript again.

Point 5: The novelty statement seems weak in its present form. Please improve the explanation of contributions made by this paper.

Response 5: We have expanded the section as follows:

“It is assumed that knowledge of setting-specific differences may be central to health systems' efforts to maintain a healthy workforce.”

Point 6: The selection of participants was random or did the authors use a convenience sampling technique? Needs clarification.

Response 6: Thank you for bringing it to our attention that we can describe the selection of participants in more detail. We have added the following sentence for this purpose:

“In this process, nurses could volunteer to participate by contacting the person in charge, and thus the selection was random.”

Point 7: Where were the scale items of constructs borrowed or modified from? Provide sources.

Response 7: The anonymized transcripts were analyzed according to the method of structuring content analysis. The chosen method is popular especially in German-speaking countries. In order to bring it closer to international readers, we have now added the following sentence:

“The method is comparable with the framework method to analyze qualitative data [33].”

With following source:

Gale, N.K.; Heath, G.; Cameron, E.; Rashid, S.; Redwood, S. Using the framework method for the analysis of qualitative data in multi-disciplinary health research. BMC Med. Res. Methodol. 2013, 13, 117, doi:10.1186/1471-2288-13-117.

Point 8: Please avoid heading after heading with nothing in between. Instead, provide at least a small paragraph to state the contents under each heading and sub-headings.

Response 8: We have checked the complete manuscript again accordingly and adjusted it where necessary.

Point 9: Conclusions are in good order.

Response 9: Thank you very much for this review.

Reviewer 2 Report

This qualitative study addresses an important issue in healthcare and although generalisations are not easy, work-related health burdens are of cardinal importance for health systems aiming at maintaining a motivated and not demoralised health-force

Suggestion:

Title: "Subjective" can be omitted since the nature and methods of the study imply it.

Abstract: Results - First sentence "the experiences.....care setting"- rephrase, it is unclear

Author Response

Thank you very much for pointing out the potential of our manuscript and the opportunity to revise it based on your comments. Your comments helped us a lot to substantially improve the quality of our manuscript. Please find our point-by-point answers in the attached document. Changes were marked up in the manuscript using the “Track Changes” function.

Response to Reviewer 2 Comments

Point 1: Title: "Subjective" can be omitted since the nature and methods of the study imply it.

Response 1: Thank you for this constructive suggestion. We have changed the title accordingly.

Point 2: Abstract: Results - First sentence "the experiences.....care setting"- rephrase, it is unclear

Response 2: Thank you for your suggestion. We have rephrased the sentence to:

“The experiences of nurses highlighted that health-related burdens have varying manifestations in different care settings.”

Reviewer 3 Report

Dear Authors,
In this version of the publication, both the content and the choice and use of research tools are correct. Undoubtedly, the research cross-sections and the description of the qualitative study can be interesting for readers.
However, I feel somewhat unsatisfied with the formulation of the Conclusions of your publication, which in my opinion are somewhat blurred. However, I am aware of the limitations, including time, in carrying out the study.
Overall my assessment is positive.

Author Response

Thank you very much for pointing out the potential of our manuscript and the opportunity to revise it based on your comments. Your comments helped us a lot to substantially improve the quality of our manuscript. Please find our point-by-point answers in the attached document. Changes were marked up in the manuscript using the “Track Changes” function.

Response to Reviewer 3 Comments

Point 1: In this version of the publication, both the content and the choice and use of research tools are correct. Undoubtedly, the research cross-sections and the description of the qualitative study can be interesting for readers.

Response 1: Thank you very much for pointing out the potential of our manuscript and the opportunity to revise it based on your comments. Your comments helped us a lot to substantially improve the quality of our manuscript.

Point 2: However, I feel somewhat unsatisfied with the formulation of the Conclusions of your publication, which in my opinion are somewhat blurred. However, I am aware of the limitations, including time, in carrying out the study.

Response 2: We have rephrased the conclusions and hope that they are now written more clearly.

Point 3: Overall my assessment is positive.

Response 3: Thank you very much for that. We are pleased that you rate our work so positively.

Reviewer 4 Report

Please see my comments in the attached document.

Author Response

Thank you very much for pointing out the potential of our manuscript and the opportunity to revise it based on your comments. Your comments helped us a lot to substantially improve the quality of our manuscript. Please find our point-by-point answers in the attached document. Changes were marked up in the manuscript using the “Track Changes” function.

Response to Reviewer 4 Comments

Point 1: Abstract: please clarify what it means by guideline-based? Which guideline was used to conduct the interviews?

Response 1: Thank you very much for this comment. With the wording we wanted to express that we used an interview guide. We apologize for the linguistic inaccuracy and have replaced the phrase guidline-based with semi-structured.

Point 2: Abstract: The description of the methods doesn’t match the interview guide (page 3). The interview was divided into five topic blocks. However, the abstract only includes four topic blocks. Professional career/career aspiration is missing. 

Response 2: We have added the missing topic block according to your recommendation.

Point 3: Introduction, para 2, what is social code XI §14? It would be helpful for international audiences who are unfamiliar with the context if the authors could explain what social code XI §14 is about.

Response 3: Thank you for bringing this to our attention. We have changed the section as follows:

“According to the social code XI §14, which defines the term "need for care" for the purpose of receiving benefits from the statutory long-term care insurance, persons are entitled if they have health-related impairments of their autonomy or abilities and therefore require help from others [6].”

Point 4: Participants: Clarify what professional nurses mean? Are they registered nurses who are licensed? Also, did the authors consider years of work experience as an inclusion criterion? How would this affect their subjectively experienced work-related burdens?

Response 4: We have made an addition in the introduction to the career access pathways as a professional nurse as follows:

“Until January 1, 2020, there were two types of nursing apprenticeships in Germany to enter the market as a professional nurse: health care nursing, trained in acute care hospitals and outpatient care services, and geriatric care nursing, trained in geriatric inpatient care facilities or outpatient care services for the elderly [15,16].”

Thank you for drawing our attention to the fact that work experience could be an interesting aspect in the selection of individuals. Since we primarily aimed for a comparison of settings, differences in work experience did not provide us with relevant results in relation to the research question.

Point 5: Table 1 shows the nurses in outpatient care service were much older than other nurses. Were they purposefully selected? What is the average age of nurses in the three settings in Germany?

Response 5:

We have described sample selection in more detail as follows:

„In this process, nurses could volunteer to participate by contacting the person in charge, and thus the selection was random.”

Unfortunately, we are not aware of sociodemographic data on nurses in different settings, as the common German reporting system analyzes data by professional affiliation.

Point 6: Data collection: what new issues arose in the setting of inpatient care? Please clarify why an additional interviewee was needed.

Response 6: Thank you for pointing out that we can make this even clearer. We have added the words "health-related".

Point 7: Discussion: the authors did a good job summarizing the results; however, it remains unclear how the findings would support the development of health promotion programs for nurses? Are there any programs currently existing? What are the goals and outcomes of such programs? What is lacking? What programs are in substantial need?

Response 7:

Thank you for bringing this to our attention. Our study is part of a larger project funded by the German Ministry of Health to get answers to your questions. We tried to make it more explicit that we are building an initial data base in this exploratory study and made the following change in the project description for this purpose:

“The project aims to implement and evaluate workplace health promotion for employees in care settings who are exposed to particular physical or psychological demands. This study belongs to the first project phase (A): Identification of relevant health problems.”

We have also made the following addition to the conclusions:

“To positively influence this with target group-specific health promotion programs, behavioral training for nurses, including raising awareness of violence, on the one hand, and reducing work demands in combination with employee-oriented changes in organizational culture, on the other, could be beneficial.”

Round 2

Reviewer 4 Report

The revised version looks good. 

Author Response

We would like to thank you once again for reading our manuscript with care and again making suggestions for improvement.
